# Click Chemistry Enabling Covalent and Non-Covalent Modifications of Graphene with (Poly)saccharides

**DOI:** 10.3390/polym13010142

**Published:** 2020-12-31

**Authors:** Hu Li, Raffaello Papadakis

**Affiliations:** 1Shandong Technology Centre of Nanodevices and Integration, School of Microelectronics, Shandong University, Jinan 250101, China; lihu.goodluck@gmail.com; 2Department of Materials Science and Engineering, Uppsala University, 751 21 Uppsala, Sweden; 3TdB Labs AB, Uppsala Business Park, 754 50 Uppsala, Sweden; 4Department of Chemistry, Uppsala University, 751 20 Uppsala, Sweden

**Keywords:** graphene, (poly)saccharides, click chemistry, hydrophilicity, click grafting, CRMs

## Abstract

Graphene is a material with outstanding properties and numerous potential applications in a wide range of research and technology areas, spanning from electronics, energy materials, sensors, and actuators to life-science and many more. However, the insolubility and poor dispersibility of graphene are two major problems hampering its use in certain applications. Tethering mono-, di-, or even poly-saccharides on graphene through click-chemistry is gaining more and more attention as a key modification approach leading to new graphene-based materials (GBM) with improved hydrophilicity and substantial dispersibility in polar solvents, e.g., water. The attachment of (poly)saccharides on graphene further renders the final GBMs biocompatible and could open new routes to novel biomedical and environmental applications. In this review, recent modifications of graphene and other carbon rich materials (CRMs) through click chemistry are reviewed.

## 1. Introduction

Graphene holds great potentials in numerous applications ranging from electronics and optoelectronics to mechanical reinforcement technologies, energy materials, sensors and actuators and many more. Properties of these materials such as aqueous stability and biocompatibility are essential, especially in the arena of biological and medical applications [1,2]. Yet, graphene is hydrophobic and irreversibly forms agglomerates or tends to restack when in water due to the strong π-π stacking interlayer interactions. This causes significant difficulties in application areas such as biomedicine and biomaterials-involving technologies [3]. To address this challenge, various functionalization approaches have been developed to enhance the dispersibility of graphene in water and other polar solvents [4]. Typically, functionalization or structure modifications can lead to new graphene-based materials with enhanced properties [5,6,7].

For instance, graphene oxide (GO) which can be obtained by chemical oxidation of graphene exhibits excellent dispersibility in various solvents [8]. Various types of functionalization, such as (hydro)silylation [9] or hydroxypropylation [10], of graphene or differential functionalization of carboxyl moieties on GO [11] and even pure geometry modification to form graphene nanoscrolls [12] have been reported as potential amendments tentatively leading to enhanced performance during dispersion. Nonetheless, many of these approaches involve reagents which are considered as non-biocompatible or toxic.

Covalent and non-covalent functionalization of graphene with mono-, di- or even poly-saccharides (PSs) has recently attracted a great deal of interest owing to the exceptionally nontoxic, hydrophilic, biocompatible, and biodegradable nature, of saccharides [13,14]. The resulting graphene derivatives show great potentials in the areas of biocomposites, biomedicine, and biosensors [15,16]. For instance, Liu et al. reported a facile and rapid green strategy to fabricate glucose functionalized graphene that is employed as a cooperative nanotemplate for both photothermal therapy and drug loading [17]. Han et al. develop a methodology to achieve graphene functionalization of pyrene-maltose as a Concavalin A biosensor, which demonstrates superior selectivity and nanomolar sensitivity [18]. It comes as no surprise that currently there is a rapidly increasing number of publications dealing with (poly)saccharide modifications of GBMs [19,20,21,22]. Yet, in order to achieve these modifications, facile, green and biorthogonal synthetic methods are required. Moreover, it is required that these synthetic methods lead to few or even no byproducts and result in high degrees of substitution. A synthetic approach fulfilling the above requirements is “click chemistry”. Up to date various click methodologies/strategies for graphene have been reported [23,24,25,26]. When it comes to (poly)saccharide modifications of graphene, there is a range of research works conducted in the last ten years which to the best of the authors’ knowledge have not been reviewed so-far. The aim of this review paper is to categorize the different click strategies that can be employed in graphene (poly)saccharide modifications. This review paper also extends to modifications of relevant carbon rich materials and compounds (CRMs), i.e., materials and compounds with a C/H ratio higher than 1/1 [27]. Important examples of CRMs are carbon nanotubes (CNTs) and polycyclic aromatic hydrocarbons (PAHs) [27], both discussed herein.

## 2. Functionalization Methodologies Involving Click Chemistry

There are various routes allowing for (poly)saccharide functionalization of graphene. One of the most prominent functionalization approaches of that type which to the best of the authors’ knowledge has not been reviewed so far, is the click chemistry approach [28]. The term click chemistry covers a range of chemical reactions all of which exhibit some important common characteristics, namely high modularity, insensitivity towards oxygen and water as well as towards the choice of solvent, high to even quantitative chemical yields, no byproducts as well as a large gain of thermodynamic enthalpy (ΔH (>20 kcal/mol)). These characteristics/requirements are met by various reactions the most prominent being the renowned copper(I)-catalyzed azide-alkyne cycloaddition (also known as CuAAC) [29]. The number of the annually published research works in the field of click chemistry is steadily higher than 1000 reflecting the high utility of click chemistry methodologies (see Figure 1). Modifications of sugars via click chemistry are common and have been in use since several years. This explains why there is a nearly stable annual number of publications in this research field during the last decade (Figure 1). Interestingly, there is an increasing number of publications pertaining to click chemistry and graphene (Figure 1) which reflects the high suitability of click chemistry for the modification of graphene and its derivatives.

The various approaches leading to polysaccharide-modified graphene can be categorized into three types (see Figure 2). In the first approach (Figure 2a), covalent graphene functionalization is achieved only on the basal plane of graphene. This typically requires a pre-modification of the desired functionality to be attached, e.g., alkyne or azide modification when CuAAC is applied as well as of the of basal plane carbon atoms of graphene (azide or alkyne modification respectively). This graphene premodification can often be a nucleophilic attack leading to epoxy ring opening in GO or a substitution of an –OH group on the basal plane of GO [30]. The second approach (Figure 2b) provides graphene functionalization at the edges and it requires similar premodifications as in approach a), the only difference being that graphene is pre-modified only at its edge carbon atoms (often through a reaction of the GO edge –COOH groups) [30]. The third approach (Figure 2c) leads to non-covalently functionalized graphene through stabilization of molecules which can be attached on the surface of graphene though weak interactions (typically through π-π interactions).

These molecules can bear a functionality which is desired for a specific application. The role of click chemistry here is to enable covalent attachment of this functionality on a molecule which shows proven affinity to the π-conjugated surface of graphene, e.g., a polycyclic aromatic hydrocarbon (PAH) [31].

Various complementary methods to click chemistry are known (see Figure 3). In many occasions, functionalization of graphene or GO is required prior to click chemistry. This is typically achieved through functionalization of GO edge-lying groups such as –COOH or –OH groups, or basal plane groups (mostly epoxide groups). For instance, in a CuAAC methodology one should first alkyne- or azide- functionalize GO, then azide- or alkyne- functionalize the target substrate (e.g., a polysaccharide) respectively to finally link the two parts via a Cu(I) catalyzed reaction. The aforementioned methodology is followed in many of the examples presented in this review work.

Direct functionalization of graphene through click chemistry can also be achieved via Diels Alder reactions or other types of cycloadditions (see Figure 3). Some relevant examples to this methodology are also included in Section 3.1.

## 3. Covalent Conjugation of (Poly)saccharides to Graphene

### 3.1. Poly- and Oligo-Saccharides

In recent years functionalization of graphene with chitosan (Structure **1** in Figure 4) has become very attractive since materials comprising these two components would exhibit a mix of the beneficial properties of graphene along with the biocompatibility, low toxicity, and biodegradability of chitosan [32]. Chitosan-based materials have attracted much interest due to their suitability for various industrial, biomedical and research applications spanning from waste water treatment [33] and oil/seawater separation systems [34] to tissue engineering [35] and carbon dioxide capturing aerogels [36]. One of the first attempts to graft chitosan on graphene came as early as 2013 by Ryu et al. and a click chemistry approach was employed for the coupling of this polysaccharide to GO [37].

In a first step, the azido moiety was introduced via a reaction of the chitosan with azido-epichlorhydrin (N-functionalization of chitosan) leading to polymer 2. GO (**3**) was then treated with propargylamine (in the presence of dicyclohexylcarbodiimide) to afford transformation of the edge carboxy groups of GO to propargyl amide groups (**4**). The two counterparts **2** and **4** were clicked together via a Cu(I)-catalyzed Huisgen 1,3-dipolar cycloaddition and chitosan-functionalized graphene oxide (**5**) was finally isolated. The described methodology enables selective functionalization of GO at its edges since the carboxy groups (lying at the edges of GO) were selectively alkyne-modified prior to the click reaction with the azide-modified chitosan.

The opposite click strategy was followed by Kabiri and Namazi in 2014 [38], to achieve a nanosized cellulose functionalized graphene. According to this approach GO was azidated whereas the polysaccharide (cellulose) was alkyne-modified (opposite to the strategy followed by Ryu et al. [37]). The reaction route, briefly described in Figure 5, leads to polysaccharide functionalization at the sheet edges of GO. In a final step reduction of GO using hydrazine leading to a final cellulose-graphene (**10**) derivative is achieved. The relevance of this synthetic strategy is high as it clearly proves the orthogonality of click chemistry [39] allowing for a post-functionalization reduction of GO without leading to an inferior degree of polysaccharide substitution. Indeed, the final functionalized material is characterized by a 23% degree of substitution by mass. It is also noteworthy that the described methodology leads to an increased hydrophilicity and dispersibility in aqueous media of the final graphene-based material and this is one of the primary reasons why (poly)saccharide graphene modifications are desired.

More recently (in 2019), click-coupling of β-cyclodextrin (β-CD) on GO was reported by Ye et al. [40]. CDs are cyclic oligosaccharides with numerous applications in environmental [41], and supramolecular chemistry [42,43,44]. They exhibit a high solubility in water yet, they retain a hydrophobic cavity interior, thereby they can efficiently trap hydrophobic molecules in aqueous solutions. This property renders CDs valuable candidates for nanocarrier systems based on carbon rich materials (CRMs), such as graphene or GO [40]. In the work by Ye et al., clicking a β-CD on graphene was made possible through an interesting approach involving click chemistry.

In a first step β-CD was partly oxidized to ox-β-CD with use of sodium periodate [45] (Figure 6, ox-β-CD: **11**). In another step, folic acid-functonalized with a maleimide unit was reacted with ox-β-CD to afford through condensation between an aldehyde unit of ox-β-CD and the folic acid amino group, scaffold (**12**). Finally, **12** was clicked on GO via a Diels-Alder click reaction to yield the GBM **13** (Figure 6) [46]. Notably, maleimide is a well-known dienophile which has been involved in Diels-Alder click reactions with a wide range of substrates, spanning from small polycyclic aromatic hydrocarbons (e.g., anthracene) [47] to CRMs, including CNTs and GBMs [48]. Remarkably, CRMs can act both as dienes and as dienophiles in Diels-Alder click reactions [49].

An alternative method allowing for the functionalization of graphene with a polysaccharide was recently (2020) described by Huang et al. [50]. In this approach, chitosan (**1**) after N-propargylation (**14**) was reacted to *para*-nitrobenzyl azide (**15**) by employing CuAAC click chemistry (Figure 7). The chitosan scaffold was further modified to the corresponding benzene diazonium tetrafluoroborate before it was finally coupled to GO to yield a chitosan-modified graphene (**19**). This approach is useful when a CuAAC click chemistry on a GBM is not directly feasible. The reactivity of aryldiazonium salts towards graphene [51,52,53] is well-known and it enables the final fixation of the pre-clicked polysaccharide scaffold. In terms of properties, it was found that the graphene-based final material exhibited an improved antimicrobial activity when compared to chitosan itself [50].

In this paragraph, recent examples of click chemistry-enabled modifications of graphene with oligo- and poly-saccharides were reviewed. Different approaches allowing for the functionalization of graphene either at its edges or at the basal plane were presented. Huisgen 1,3-dipolar cycloaddition reactions as well as Diels Alder click reactions were employed in most of the cases. In the next paragraph focus is placed on mono- and di- saccharides.

### 3.2. Mono and Di-Saccharides

One of the earliest attempts to functionalize graphene with small sugars (mono- or di-saccharides) using a click chemistry methodology was that by Namvari and Namazi [54]. In a first step, graphene oxide (GO) was azide-modified through simple reactions, and then it was functionalized with a variety of mono- and di-saccharides via Cu(I) catalyzed Huisgen 1,3-dipolar cycloaddition reactions [29,55]. The saccharides (galactose, maltose, glucose and mannose) were alkyne-modified in a separate step. The resulting family of these sugar-graphene conjugates which the authors called “sweet graphene” [54] exhibited interesting properties such as very good dispersibility in water as well as high stability. What was shown is that the same type of functionalization can be achieved both at the basal plane and at the edges of graphene by modifying the “azidation” step. By reacting GO with NaN_3_, click is directed towards the epoxy groups (basal plane). On the other hand, by first reacting GO with 1,3-diazidopropan-2-ol (after activating the terminal carboxylic acid units (edges) through reaction with SOCl_2_), the click reactions with the alkyne-sugars finally yield an edge-modified GO. This approach indicates the versatility of the click-methodology enabling two functionalization modes (basal plain or edges functionalization) while retaining the same substrate.

The same research group later on reported on an extension of their first work on “sweet-graphene” in which they employed two different CuAAC click-reactions in order to achieve basal plane and edges functionalization of GO with glucose. (see Figure 8) [56]. In a similar fashion as before, this required the pre-alkyne modification of GO at the edge -COOH entities or at the -OH basal plane entities (Figure 8). Coupling reactions of each of these pre-modified GOs with azido-ethylene glucose (structure **22**
Figure 8) in the presence of CuSO_4_ and ascorbate (reducing agent) resulted in glucose-functionalized GOs. What is interesting especially with the basal plane modification is that a dendrimer-like structure is achieved since the GO precursor involved entities with three alkyne groups each. This allowed coupling to three units of glucose through the applied CuAAC approach. The resulting GBM is characterized by enhanced hydrophilicity [56]. The hydrophilic graphene nanosheets were further combined to Fe_3_O_4_ nanoparticles and the superparamagnetic properties of the final materials were analyzed [56]. The efficient and stable deposition of the Fe_3_O_4_ nanoparticles is clearly associated to the hydrophilicity of the synthesized materials through click chemistry.

In this paragraph substitution reactions of graphene with mono- and di- saccharides were reviewed and the required preparations of the graphene and saccharide precursors were discussed. A Cu(I) catalyzed Huisgen 1,3-dipolar cycloaddition was identified as the click method mostly used for this type of graphene covalent modifications. This is presumably because of the ample existing knowledge on the azide- or alkyne-modifications of mono- and di-saccharides [28]. In the next paragraph, click-assisted non-covalent graphene modifications are discussed.

## 4. Non-Covalent Conjugation of (Poly)saccharides to Graphene

Another way to modify graphene is through the so-called non-covalent functionalization approach. In this case, the functionalities are not bound to graphene by means of a covalent bond but instead they are typically bound to a compound which can stay on the surface of graphene by means of relatively weak interactions, e.g., through π-π interactions between a PAH and graphene. This strategy is often preferred when it is desired that the electrical and electronic properties of graphene are retained after functionalization and this can be ensured by means of methodologies which retain intact the fully π-conjugated honeycomb structure of graphene [31,57]. Ιn this section, non-covalent approaches allowing for saccharide functionalization of graphene are reviewed.

Kaminska et al. reported in 2012 on a click-chemistry assisted method to non-covalently functionalize GO [58]. By using an azido-tetrathiafulvalene (az-TTF: **25**) scaffold which they initially non-covalently functionalized GO and in one step they achieved reduction of GO to rGO merely using sonication. The reducing capacity of the az-TTF is such that allowed for the GO to rGO reduction. The result is a GBM involving non-covalent az-TTF-functionalization. Moving a step further it was possible to apply a CuAAC chemistry to tether a monosaccharide unit (mannose) which was pre-alkyne modified. The final material was thus a modified graphene involving a monosaccharide prepared in two steps (structure **28**). What is even more interesting is that the mannose-TTF units can reversibly be detached form the graphene surface merely by applying the tetracationic cyclophane cyclobis(paraquat-p-phenylene): **29** which efficiently binds mannose-TTF. The process was nicely monitored by means of Atomic Force Microscopy (AFM) (see Figure 9C).

Subsequently, Zhang et al. [59] reported on graphene modified materials useful as fluorogenic sensors for the recognition of specific intercellular glycoprotein receptor interactions. The authors characterized this GBM as “2D glycosheet”. Click chemistry enabled the preparation of these materials (see Figure 10). The synthetic approach followed, involved two different N-acetyl hexosamine rhodamine B derivatives (compounds **32** and **33**) varying on the type of monosaccharide employed (N-acetyl galactosamine in case of **32** and N-acetyl glucosamine in case of **33**). Alkyne-modified rhodamine B was attached to each azido-monosaccharide through a CuAAC reaction. The π-conjugated nature of the rhodamine matches well the partly π-conjugated honeycomb structure of GO. Assembly of **32** and **33** on GO occurs efficiently leading to the glycosheet materials **34–35** respectively [59]. What is very interesting in this system is that rhodamine B plays both the role of the π-conjugated entity allowing for the stacking/stabilization on GO and the role of the fluorescent group which is the essential feature of this fluorogenic system.

Saccharides tethered on π-conjugated molecules (e.g., PAHs) through click chemistry can also lead to non-covalent functionalizations of CRMs other than graphene. Assali et al. recently reported on the development of modified MWCNTs with sugar-based amphiphiles tethered on tetrabenzo[a,c,g,i]fluorene (TBF) (see Figure 11) [60]. The π-π stacking interactions between the aforementioned aromatic entities and the surface of MWCNTs enabled the preparation of this interesting composite material involving CΝΤs and sugar entities. The tethering of the sugars was achieved through an oligo ethylene glycol bridge with a terminal azide which via a CuAAC reaction with alkyne-modified TBF resulted in various sugar amphiphiles (example structure **36**). Amphiphiles like **36** efficiently bind the surface of MWCNTs non-covalently through π-π stacking interactions forming composites similar to **37** (Figure 11). In terms of applications, the as formed aggregates were found to be involved in specific lectin-ligand interactions in an analogous fashion to glycoconjugates on a cell-membrane [60,61].

Wu et al. [62], some years earlier, reported on CNTs functionalization with glycodendrimers involving a pyrene tail which exhibits high aptitude to attach non-covalently the surface of CNTs (as well as other CRMs, e.g., graphene). The glycodendrimers were prepared through a click chemistry-assisted methodology leading to triazole repeated binding motifs. This way, pyrene branch-terminating monosaccharides are formed. The glycodendrimers by Wu et al. have been proposed for protective coatings and they are capable to moderate the cytotoxicity of CNTs when bound to their surface [62].

Pyrene among other PAHs is an indispensable part of many non-covalent functionalizations to graphene, CNTs and other CRMs [63,64,65]. This fact is reflected by the large number of published works in this research field pertaining to GBM functionalization strategies which involve derivatized pyrenes [5,6,63,64,65]. There is also a significant subgroup of the aforementioned research field involving pyrene sugar-modifications enabled by click chemistry which could lead to interesting composites when combined with graphene or other CRMs. Furthermore, studying such PAH modifications with saccharides constitutes an important step prior to applying covalent functionalization strategies on graphene. Indeed, modelling graphene modifications with the use of variously sized PAHs has been identified as an important research approach [66,67]. Some characteristic selected examples of this class of compounds are depicted in Figure 12 [68].

## 5. Covalent Click Chemistry Approaches Applied to Other CRMs

In this paragraph, focus is placed on click approaches allowing for covalent functionalization of CRMs other than graphene. One of the most prominent allotropes of carbon which during the last decades has attracted a great deal of interest is CNTs [72]. CNTs constitute a well-studied type of CRMs in terms of their outstanding physicochemical properties. Numerous applications of CNT-involving materials have been proposed to date spanning from optoelectronic to environmental applications [72,73,74,75,76]. One of their important characteristics is their very high aspect ratio (length to diameter ratio can be higher than 10^6^ [73]). A vast number of described functionalization methodologies for CNTs is known to date [72]. Yet, a common problem for many applications is the insolubility of CNTs in any organic solvent or water [28,72]. Many attempts have been made towards resolving this problem and click chemistry approaches have demonstrated a great role towards this endeavor [77,78]. Click chemistry offers various covalent functionalization possibilities for CNTs [28]. Various oligo- and poly-saccharides have been employed up to date as hydrophilic units which increase the solubility and dispersibility of CNTs in various solvents. In the previous paragraph some examples of non-covalent functionalizations of CNTs were reviewed yet, here focus is placed on covalent functionalization approaches as these approaches could lead to soluble CNT-based materials.

Guo et al. [79] reported on the functionalization of CNTs with β-CD through a CuAAC methodology using CuI-DBU and DMF as a solvent. (Figure 13A). A first pre-alkyne-modification of CNTs was required. This was achieved through a reaction of CNTs with *p*-amino-O-propargylphenol and isoamylnitrite to afford product **42** (Figure 13A). The side-wall alkyne-modified CNTs (**42**) efficiently reacted with azide-functionalized β-CD (**43**) to form β-CD-CNTs (**44**). The method described led to a new type of functional material. As an application, the binding ability of the β-CD-CNT fluorescence spectroscopic study was conducted towards quinine which was found to efficiently enter the β-CD cavity of 43 [79]. (It is worth mentioning that quinine/β-Cyclodextrin constitutes an important modelling system for simulating enzyme−substrate interactions [80]). Notably, this approach is similar to the method employed by Ye et al. [40] (see Figure 6) pertaining to GO yet in the latter case another pre-modification of the CRM was required prior to click.

Click strategies other than the CuAAC chemistry allowing for the grafting of CNT-involving materials with polysaccharides have also been reported. Yadav et al. for example reported on the functionalization of multiwalled carbon nanotubes (MWCNTs **45**) with chitosan (**1**) [81]. The synthetic strategy employed involved a first derivatization (through amidation) of chitosan with a terminal azide (see Figure 13B) which was then directly clicked on the surface of MWCNTs by means of a [2 + 1] nitrene cycloaddition to afford product **46**. The latter reaction has recently been applied on CNTs [82,83,84] with success enabling grafting of various functionalities on CNTs. Noteworthy [2 + 1] nitrene cycloaddition on CNTs leads to derivatives which retain π-conjugation rendering this method very attractive for applications in optoelectronics.

From a sugar-CNTs-functionalization perspective, what makes the click reaction by Yadav et al. attractive, is the fact that only one derivatization of the polysaccharide with a linker encompassing an azide-terminus is required. The CNTs are then used directly, i.e., without any pre-derivatization. The reaction is brought about at temperatures typically higher than 150 °C in solvents like N-methyl-2-pyrrolidone (NMP). The reported degree of substitution on MWCNTs in the final product (**47**) was as high as 1.8 chitosan polymeric chains every 1000 carbon atoms of MWCNTs. When it comes to the properties of the final material it was shown that it exhibits enhanced mechanical properties and antimicrobial activity when compared to the parent materials.

In this paragraph it has been shown that click-chemistry approaches are also suitable for CRM-modifications with sugars (CRMs other than GBMs). The techniques used are the same as described in corresponding graphene modifications. Nonetheless, different pre-modification methods are applied depending on the CRM-substrate.

## 6. Future Perspectives

To the best of the author’s knowledge, very few examples on click-sugar modified CRMs other than CNTs and GBMs can be found. There are various examples involving PAHs (for example, see Figure 12 and corresponding discussion) however, the field is open to many more combinations of CRMs and (poly)saccharides. The increasing tendency of annual number of published works pertaining to graphene modifications using click chemistry (see Figure 1) signifies a bright future for this research field which can bring together the world of sugars and that of graphene chemistry. Of particular interest are Diels Alder click-reactions, which are becoming more and more popular. On the other hand, CuAAC click-reactions are very attractive since both azide- and alkyne- modifications of sugars are facile and have been long studied for other purposes (e.g., in bioconjugation).

## 7. Conclusions

This paper provides a review on the recent research pertaining to the modifications of graphene and other CRMs with (poly)saccharides enabled by click chemistry. Emphasis was given to click chemistry approaches which have been shown to allow for either covalently or non-covalently graphene functionalizations. The review expands to CRMs like CNTs and includes a range of examples and potential applications of the (poly)saccharide modified materials. In most cases reviewed, it was shown that (poly)saccharide modifications increased the hydrophilicity and dispersibility of CRMs in water or other solvents. Moreover, the materials thus produced exhibit multifunctional behavior and are proposed as important candidates for various new applications.

## Figures and Tables

**Figure 1 polymers-13-00142-f001:**
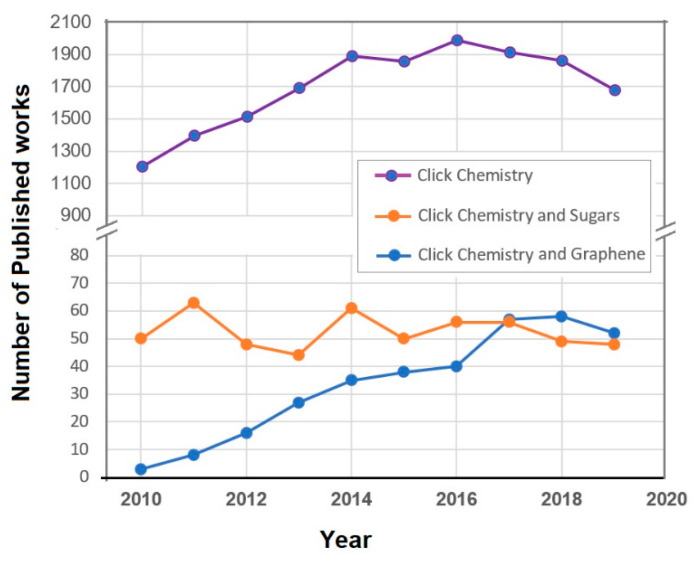
Plots depicting the number of published works in three research areas relevant to this review paper for the last ten years. Keywords used: “Click Chemistry”, “Click Chemistry” and “Sugars” and “Click Chemistry” and “Graphene”. Searched through: Web of Science. Search performed on on 26 November 2020.

**Figure 2 polymers-13-00142-f002:**
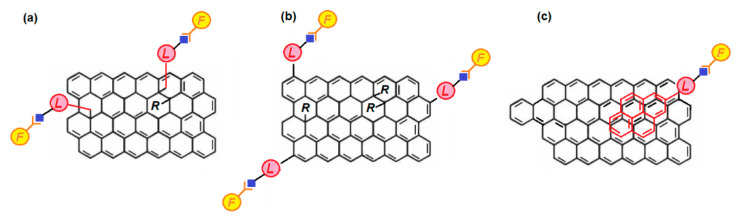
Covalent functionalization of graphene enabled through click chemistry in the basal plane (**a**) and at the edges (**b**). Non-covalent modification of graphene through π-π stacking interactions of a functionalized pyrene through click chemistry (**c**) (**L** corresponds to a linker, **F** to a functional group attached through a click approach. ∟ and ◆ correspond to the two complementary chemical parts prone to undergo a click-coupling; **R** random graphene substituent).

**Figure 3 polymers-13-00142-f003:**
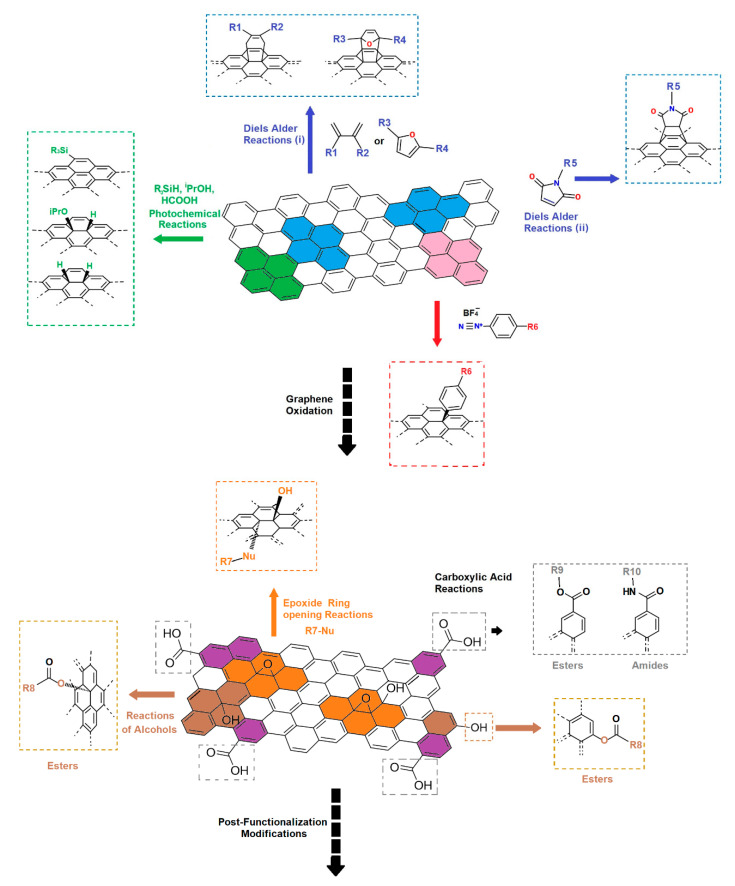
Various functionalization reactions of graphene (upper part) and GO (lower part) relevant/complementary to click chemistry. (R1 to R10 represent various substituents; Nu corresponds to a nucleophile).

**Figure 4 polymers-13-00142-f004:**
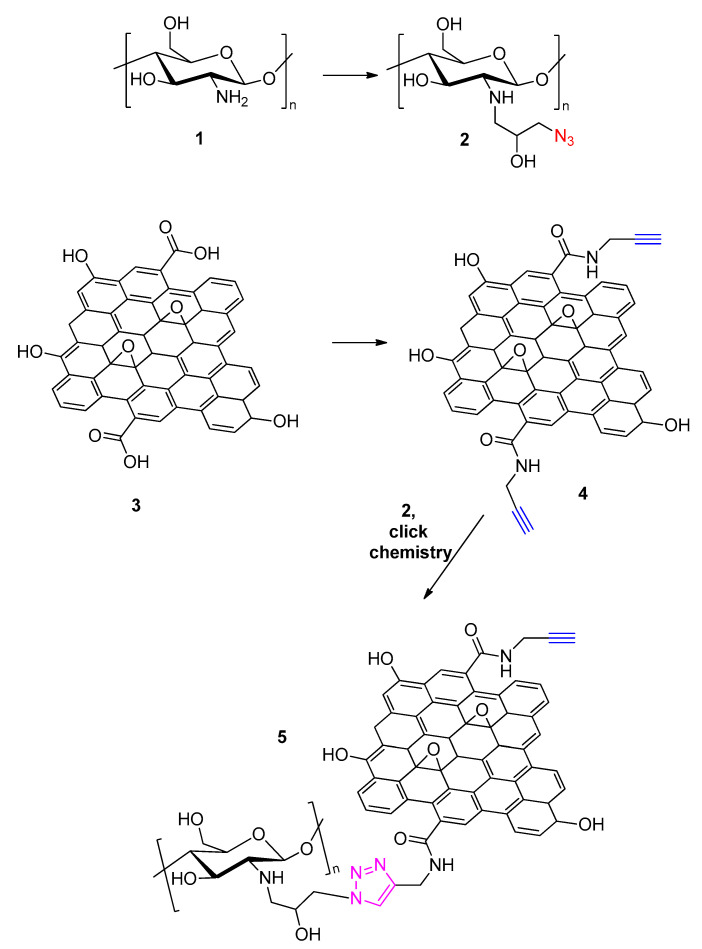
Synthetic route followed by Ryu et al. [37] for the preparation of chitosan functionalized graphene (5).

**Figure 5 polymers-13-00142-f005:**
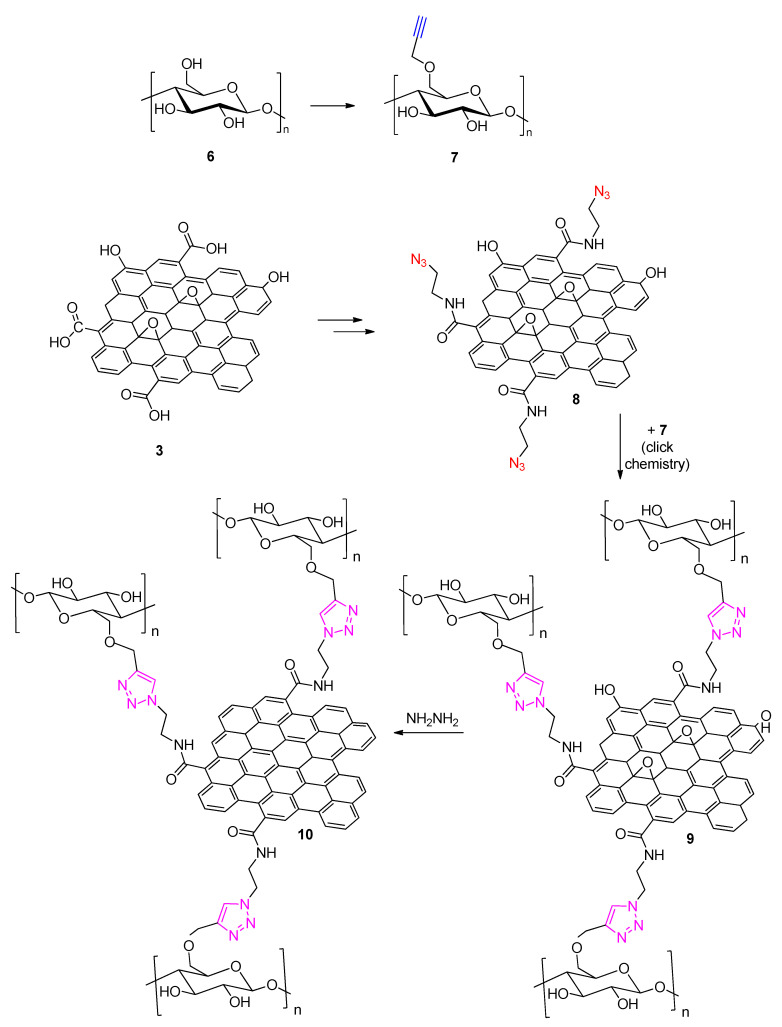
Synthetic strategy employed by Kabiri and Namazi [38] for the preparation of cellulose-modified graphene.

**Figure 6 polymers-13-00142-f006:**
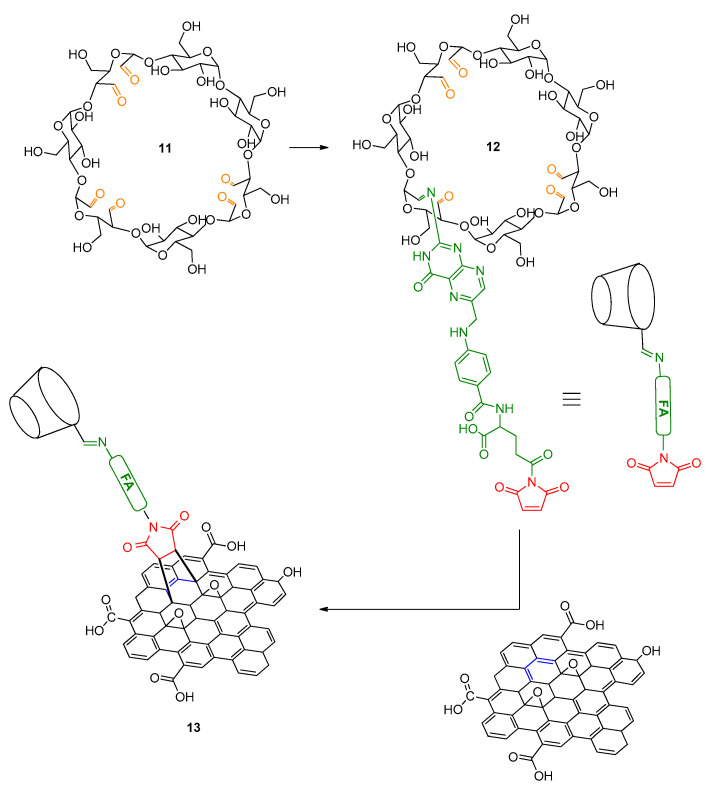
Synthesis of click-coupled β-CD on graphene by Ye et al. [40].

**Figure 7 polymers-13-00142-f007:**
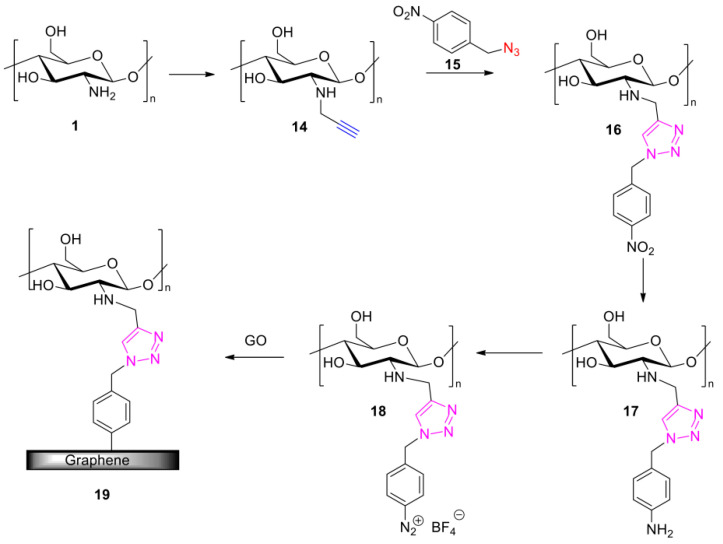
CuAAC Click assisted synthesis of chitosan-modified graphene by Huang et al. [50].

**Figure 8 polymers-13-00142-f008:**
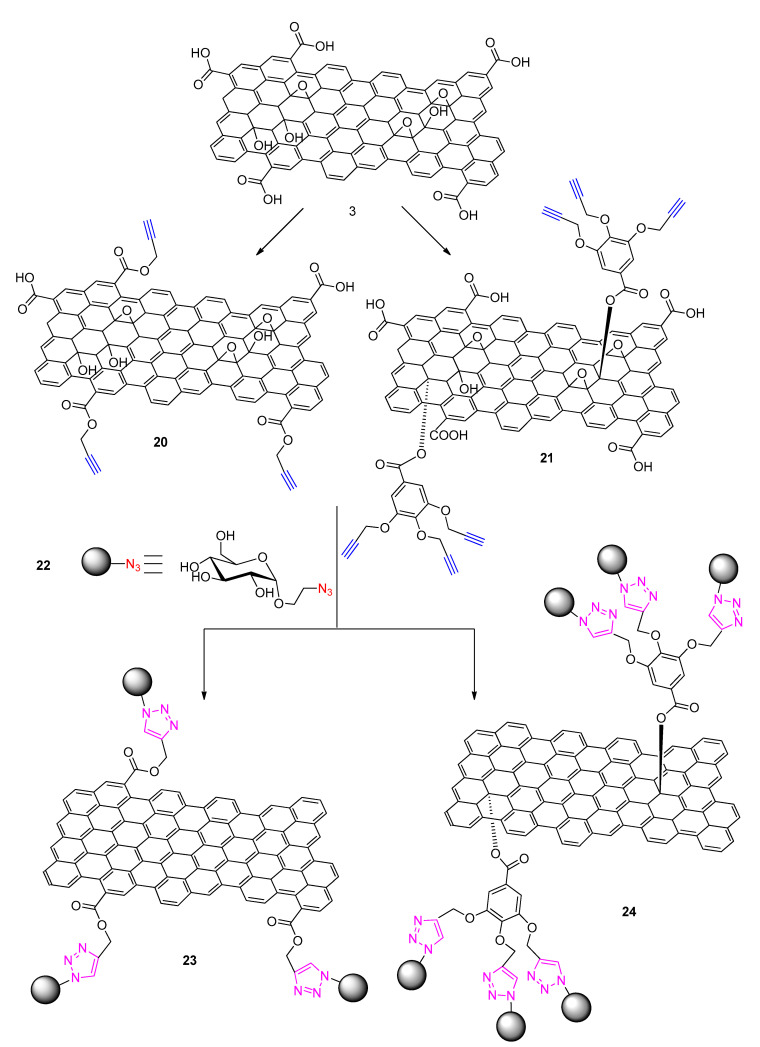
The two click-assisted synthetic routes to graphene glucose functionalization at the edges (product **23**) and at the basal plane (**24**) by Namvari and Namazi [56].

**Figure 9 polymers-13-00142-f009:**
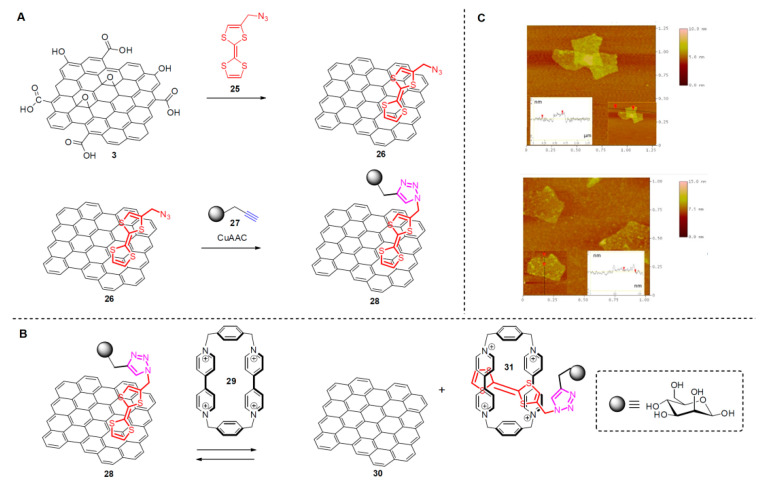
(**A**) Non-covalent functionalization of r-GO with TTF-mannose (**B**) Reversible removal of mannose-TTF from graphene and; inset depict the structure of the monosaccharide mannose (**C**) AFM images before (top) and after (bottom) immersing the r-GO-TTF-mannose in a solution of cyclobis(paraquat-p-phenylene). The AFM images are reproduced with permission from Kaminska et al. [58].

**Figure 10 polymers-13-00142-f010:**
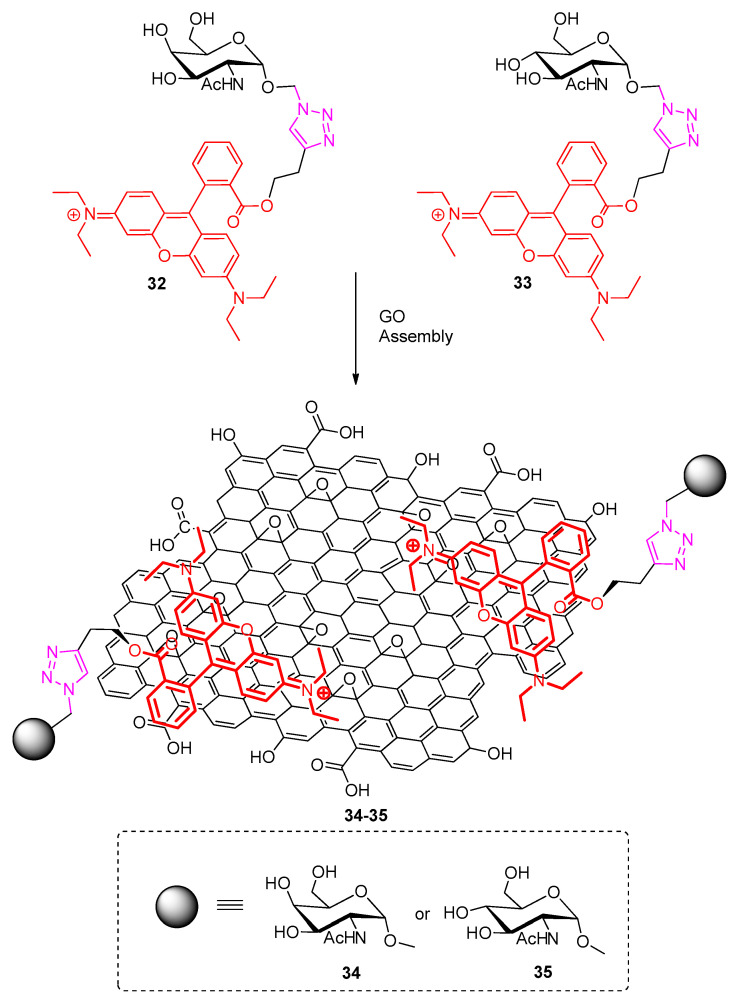
Click chemistry enabled preparation of the 2D glycosheet materials by Zhang et al. [59].

**Figure 11 polymers-13-00142-f011:**
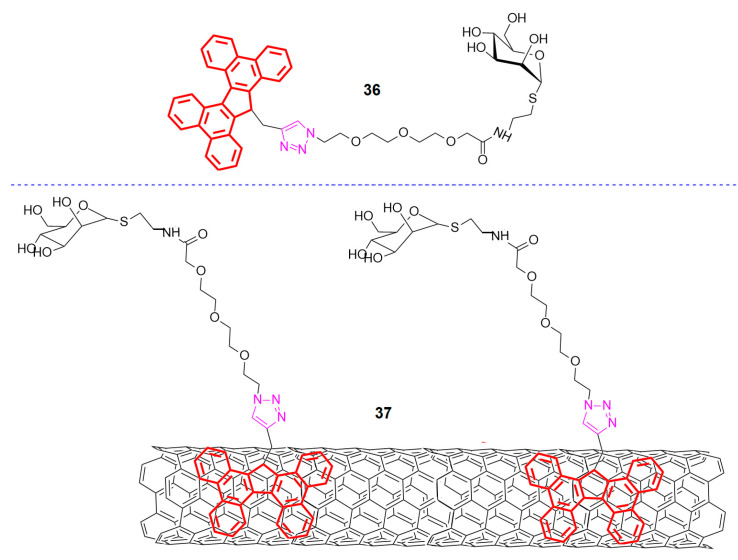
A sugar amphiphile and its aggregate with MWCNTs by Assali et al. [60].

**Figure 12 polymers-13-00142-f012:**
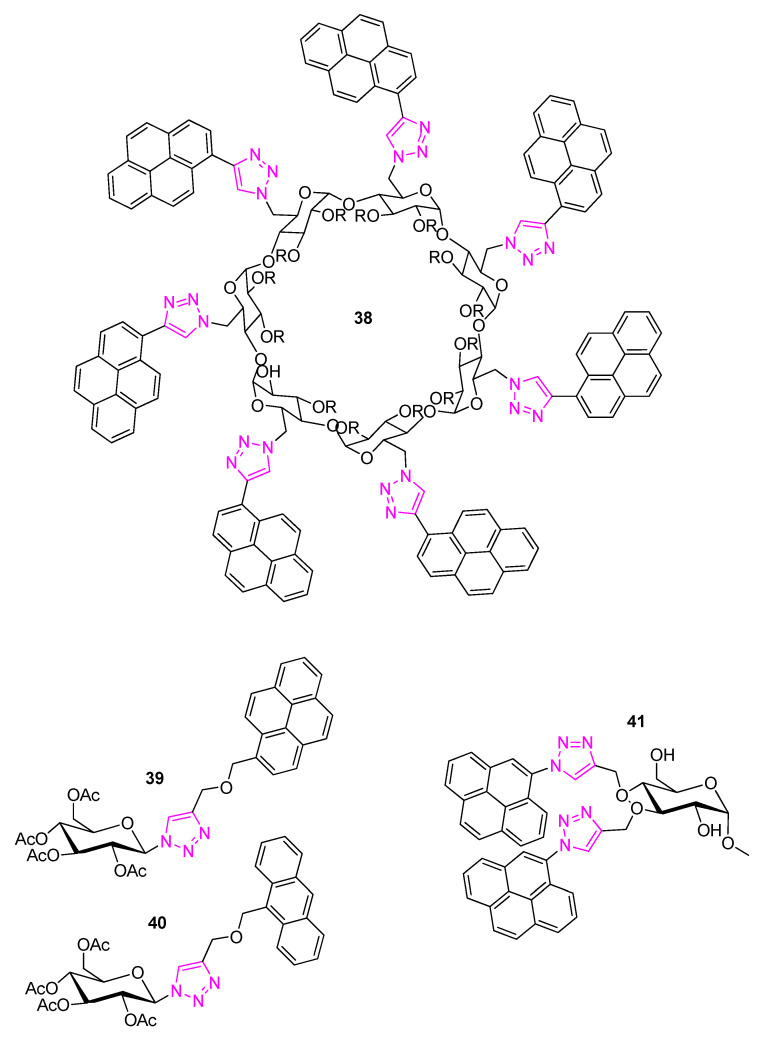
Examples of a saccharide modified pyrene derivatives. A pyrenyl β-cyclodextrin **38** by He et al. [69], pyrenyl (**39**) and anthracenyl (**40**) glucose derivatives by Thakur et al. [70] and glycoprobe by He et al. [71].

**Figure 13 polymers-13-00142-f013:**
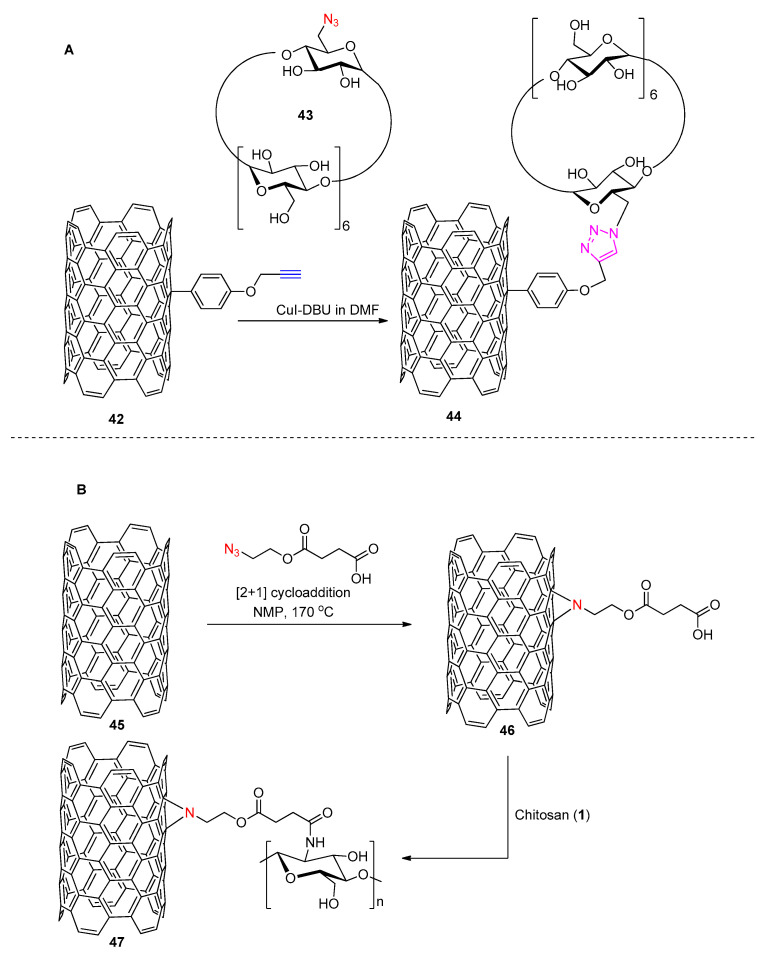
(**A**) Synthetic procedure used for the coupling of β-CD on SWCNTs by Guo et al. [79] and (**B**) synthetic route followed for the chitosan-functionalization of MWCNTs by Yadav et al. [81].

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
