# Peer review of "Click Chemistry Enabling Covalent and Non-Covalent Modifications of Graphene with (Poly)saccharides"

_polymers, 2020, doi:10.3390/polym13010142_

Round 1

Reviewer 1 Report

This review manuscript offers a brief introduction of click chemistry in the modification of graphene and other carbon nanotubes with (poly)saccharides. The cite references are fresh and extensive. However, the context catalogue is not so relevant and the summary based on the references is not so comprehensive. For examples, click reaction includes several sorts such as alkynyl-azido cyclo-addition, thio-alkene addition, malaimide-alkene cyclo-addition and so on; the application of modified graphene with (poly)saccharides is not well introduced; carbon nano-structures other than graphene are also included. Beside those academic problems, the writing should be greatly improved since there are too many complicated long sentences, unclear expression, improper usage of phrases and type-errors, making one reader to instantly follow what this manuscript should express. Thus, this manuscript should be re-considered after major revision and great improvement. Some comments and suggestions are given below.

  • Abstract: “remain” is an intransitive verb and “its insolubility and poor dispersibility” should not be attached behinds. As well, “its insolubility and poor dispersibility” is “two” but not “one” drawback. Chemical modification of graphene and other nano-carbons includes various routes. Click chemistry is one important route. The phrase of “descent dispersibility” seems a little strange. The word of “biocompatible” should be “biocompatibility” and “chapter” is used as “review”. What is “carbon rich materials”? Are they carbon nano-meterials, such as CNT and C60? If “modifications enabled by click chemistry and applied on graphene” is revised as “modifications of graphene through click chemistry”, the expression could be concise.
  • Line 43 and 44: what kinds are the “factors”? “Factors” can not be “biocompatible” and “toxic”.
  • Line 55-57: this sentence is a typical example of implicated expression. As well, “are scarce” should be “were scarce” since click chemistry has been established for more than two decades.
  • Line 71 and 72: reaction mechanism other than thermodynamic factor can dominate if by-products would be produced.

Please check the whole manuscript and modify other similar problems, some of which have been annotated with color mask or color underlines.

  • Reference citation: With the information of Ref 30, the reviewer failed in picking it out through Web of Science and Scopus. Additionally, Ref 30 looks like a review and this citation would be in-direct. Please cite the original literature. As for Ref 37, please check if there is any experimental data to support the statement that “the described methodology enables efficient functionalization of GO at the edges as it was chosen to alkyne-modify only the carboxy groups of GO at its edges”. Also, please check other references for improper citation.

Author Response

Comment: This review manuscript offers a brief introduction of click chemistry in the modification of graphene and other carbon nanotubes with (poly)saccharides. The cite references are fresh and extensive. However, the context catalogue is not so relevant and the summary based on the references is not so comprehensive. For examples, click reaction includes several sorts such as alkynyl-azido cyclo-addition, thio-alkene addition, malaimide-alkene cyclo-addition and so on; the application of modified graphene with (poly)saccharides is not well introduced; carbon nano-structures other than graphene are also included. Beside those academic problems, the writing should be greatly improved since there are too many complicated long sentences, unclear expression, improper usage of phrases and type-errors, making one reader to instantly follow what this manuscript should express. Thus, this manuscript should be re-considered after major revision and great improvement. Some comments and suggestions are given below. 

Our reesponse:

We would like to thank Reviewer 1 for the productive criticism on our manuscript. We have seriously taken into account the proposed changes and we hope that our manuscript is in much better shape after this revisions. We would also like to inform the reviewer that our text modifications are highlighted (yellow) and that two new Figures have been added as responses to comments by Reviewer 2. Our point-by-point responses to all comments by Reviewer 1. 

Comment 1: Abstract: “remain” is an intransitive verb and “its insolubility and poor dispersibility” should not be attached behinds. As well, “its insolubility and poor dispersibility” is “two” but not “one” drawback.

Our response: This has been taken into account and this sentence now reads:

"However, the insolubility and poor dispersibility of graphene are two major problems hampering its use in certain applications"

Comment 2: Chemical modification of graphene and other nano-carbons includes various routes. Click chemistry is one important route. The phrase of “descent dispersibility” seems a little strange. 

Our response: This has now be corrected and this sentence now reads:

"Tethering mono-, di- or even poly-saccharides on graphene through click-chemistry gains more and more attention as a key modification approach leading to new graphene-based materials (GBM) with improved hydrophilicity and substantial dispersibility in polar solvents e.g. water"

Comment 3: The word of “biocompatible” should be “biocompatibility”

Our response: We believe that this sentence is correct as it is i.e.:

"The attachment of (poly)saccharides on graphene further renders the final GBMs biocompatible"  Biocompatibility is not appropriate in this particular sentence. 

Comment 4: “chapter” is used as “review”.

Our response: This has been a careless mistake of ours. We have now corrected all the terms “chapter” to “review” throughout the whole manuscript.

Comment 5: What is “carbon rich materials”? Are they carbon nano-meterials, such as CNT and C60?

Our response: We clarify this now through the following sentence:

"This review paper also expands to modifications of relevant carbon rich materials and compounds (CRMs) i.e. materials and compounds with C/H ratio higher than 1/1.[27] Important examples of CRMs are carbon nanotubes (CNTs) and polycyclic aromatic hydrocarbons (PAHs)[27] both discussed herein."

Comment 6: If “modifications enabled by click chemistry and applied on graphene” is revised as “modifications of graphene through click chemistry”, the expression could be concise.

Our response: This has now be corrected accordingly.

Comment 7: Line 43 and 44: what kinds are the “factors”? “Factors” can not be “biocompatible” and “toxic”.

Our response: The term factors has been removed. The sentence now reads:

"Nonetheless, many of these approaches involve reagents which are considered as non-biocompatible or toxic."

Comment 8: Line 55-57: this sentence is a typical example of implicated expression. As well, “are scarce” should be “were scarce” since click chemistry has been established for more than two decades.

Our response: This has now been corrected.

Comment 9: Line 71 and 72: reaction mechanism other than thermodynamic factor can dominate if by-products would be produced.

Our response: This is a very good point. We have modified this sentence as follows:

"The term click chemistry covers a range of chemical reactions all of which exhibit some important common characteristics, namely high modularity, insensitivity towards oxygen and water as well as towards the choice of solvent, high to even quantitative chemical yields, no byproducts as well as a large gain of thermodynamic enthalpy [ΔΗ (>20 kcal/mol)]."

Comment 10:Please check the whole manuscript and modify other similar problems, some of which have been annotated with color mask or color underlines.

Our response: We have performed a careful check of our manuscript and we have identified and revised sentences which were either too long/complicated as well as sentences involving grammar or syntax errors. All changes are highlighted in yellow as mentioned.

Comment 11: Reference citation: With the information of Ref 30, the reviewer failed in picking it out through Web of Science and Scopus. Additionally, Ref 30 looks like a review and this citation would be in-direct. Please cite the original literature.

Our response: We thank Reviewer 1 for the thorough review. Indeed reference 30 is a book. This reference was provided as the part of the text to which it corresponds is: "This graphene premodification can often be a nucleophilic attack leading to epoxy ring opening in GO or a substitution of an -OH group on the basal plane of GO.[30] The second approach (Fig. 2b) provides graphene functionalization at the edges and it requires similar premodifications as in approach a), the only difference being that graphene is pre-modified only at its edge carbon atoms (often through a reaction of the GO edge –COOH groups). [30]"  

This text is a general introduction to the generic "chemistries" that GO can undergo and can be found not only in one, but in numerous publications. The book was picked not randomly. The authors happen to own this book and we believe this can nicely convey the message of the as above statement. Nonetheless, since Reviewer disagrees with this choice, we have changed reference [30] to the following review paper describing all three general routes mentioned which we believe will help the reader understand the high versatility of GO when graphene modifications are intented. We really hope that Reviewer 1 will find this as a good alternative to the book reference:

 [30] Dreyer D R, Park S, Bielawski C W, Ruoff R S. The chemistry of graphene oxide. Chemical Society Reviews, 2010;39:228–240. DOI: 10.1039/b917103g.

Comment 12: As for Ref 37, please check if there is any experimental data to support the statement that “the described methodology enables efficient functionalization of GO at the edges as it was chosen to alkyne-modify only the carboxy groups of GO at its edges”.

Our response: The sentence involved a statement by the authors (us) and we agree that there is no such a mention in Ref 37 justifying this statement.

Therefore, we agree with Reviewer 1 and we have now modified the sentence which corresponds to Ref 37 as follows:  

"The described methodology enables selective functionalization of GO at its edges since the carboxy groups (lying at the edges of GO) were selectively alkyne-modified prior to the click reaction with the azide-modified chitosan."

Comment 13:Also, please check other references for improper citation.

Our response: We have performed all the checks/changes pertaining to this comment.

Reviewer 2 Report

The manuscript titled “Click Chemistry Enabling Covalent and Non-Covalent Modifications of Graphene with (Poly)saccharides” with the Manuscript ID: polymers-1012186 by Li et al. has been reviewed. The authors have reviewed recent modifications enabled by click chemistry and their applications. A few of the concerns are listed below for the authors to be addressed.

  1. Line 21, 342: The word ‘chapter’ seems inappropriate as this manuscript is a review work. I am guessing that it is not going for a book chapter.
  2. English language (some sentences) needs correction in some places, e.g., line 57
  3. A plot to show the number of publications per year taking place in the relevant field would be of help to understand the growth of work in this particular area for the last 10 years as it is assumed that they have reviewed the papers published during the last 10 years as per their claim (line no. 60).
  4. Mentioning the ‘keywords’ used for searching and selecting the research papers which the authors have reviewed are to be mentioned. Also mention the database and duration (which year to which year) used for the search.
  5. A figure summarizing various ways to functionalize graphene would increase reader’s interest.
  6. In each section, the main conclusion should be given so that the readers can easily find the advantages/disadvantages and required improvement(s).
  7. Finally, a future direction/perspective for the improvement should be proposed.

Author Response

General Comment: The manuscript titled “Click Chemistry Enabling Covalent and Non-Covalent Modifications of Graphene with (Poly)saccharides” with the Manuscript ID: polymers-1012186 by Li et al. has been reviewed. The authors have reviewed recent modifications enabled by click chemistry and their applications. A few of the concerns are listed below for the authors to be addressed.

Our response: We would like to thank Reviewer 2 for bringing up important omissions in our first submission. We have taken in to account all comments and herein we provide responses to all comments. We would also like to inform the reviewer that our text modifications are highlighted (yellow). Our point-by-point responses to all comments by Reviewer 2 are as follows: 

Comment 1: Line 21, 342: The word ‘chapter’ seems inappropriate as this manuscript is a review work. I am guessing that it is not going for a book chapter.

Our response: The word "chapter" has been replaced by the word "review" throughout the whole manuscript.

Comment 2: English language (some sentences) needs correction in some places, e.g., line 57.

Our response: Agreeing with this comment, we have performed a thorough check on our manuscript and have implemented important changes which are all highlighted in yellow.

Comment 3: A plot to show the number of publications per year taking place in the relevant field would be of help to understand the growth of work in this particular area for the last 10 years as it is assumed that they have reviewed the papers published during the last 10 years as per their claim (line no. 60).

Our response: This is an important omission and we have now:

a) added Figure 1 (plot of annual number of publication in three research fields relevant to the review during the last decade. We include the details on our web search in the Fig. caption.

b) We have added the following text explaining the findings of Fig.1

"The number of the annually published research works in the field of click chemistry is steadily higher than 1000 reflecting the high utility of click chemistry methodologies (see Fig. 1). Modifications of sugars via click chemistry are common and have been in use since several years. This explains why there is a nearly stable annual number of publications in this research field during the last decade (Fig. 1). Interestingly, there is an increasing number of publications pertaining to click chemistry and graphene (Fig. 1) which reflects the high suitability of click chemistry for the modification of graphene and its derivatives"

Comment 4: Mentioning the ‘keywords’ used for searching and selecting the research papers which the authors have reviewed are to be mentioned. Also mention the database and duration (which year to which year) used for the search.

Our response: We have included the keywords requested by Reviewer 2 (Line 419) as follows:

"Keywords used for online search

Online research for this review paper was mainly performed via Google Scholar (https://scholar.google.com/) using the following search keywords: “Graphene + functionalization + polysaccharides + click” ; “Graphene + functionalization + sugars + click”; “CNTs + functionalization + sugars + click” and “PAHs + functionalization + sugars + click”. Years: 2008-today."   

Comment 5: A figure summarizing various ways to functionalize graphene would increase reader’s interest.

Our response: This is a very good point. We have included this as Figure 3.

Fig. Caption: Figure 3. Various functionalization reactions of graphene (upper part) and GO (lower part) relevant/complementary to click chemistry. (R1 to R10 represent various substituents; Nu corresponds to a nucleophile).

We also included a short discussion on this new Figure entry:

"Various complementary methods to click chemistry are known (see Fig. 3). In many occasions, functionalization of graphene or GO is required prior to click chemistry. This is typically achieved through functionalization of GO edge-lying groups such as -COOH or -OH groups, or basal plane groups (mostly epoxide groups). For instance, in a CuAAC methodology one should first alkyne- or azide- functionalize GO, then azide- or alkyne- functionalize the target substrate (e.g. a polysaccharide) respectively to finally link the two parts via a Cu(I) catalyzed reaction. The aforementioned methodology is followed in many of the examples presented in this review work. Direct functionalization of graphene through click chemistry can also be achieved via Diels Alder reactions  or other types of cycloadditions (see Fig. 3). Some relevant examples to this methodology are also included in paragraph 3.1."

Comment 6: In each section, the main conclusion should be given so that the readers can easily find the advantages/disadvantages and required improvement(s). 

Our response: we have added a short conclusion to each section according to the request by Reviewer 2.

Comment 7: Finally, a future direction/perspective for the improvement should be proposed.

Our response: In response to this comment, we have now included a the following paragraph

"Future perspectives

To the best of the author’s knowledge, very few examples on click-sugar modified CRMs other than CNTs and GBMs can be found. There are various examples involving PAHs (see for example Fig. 12 and corresponding discussion) however, the field is open to many more combinations of CRMs and (poly)saccharides. The increasing tendency of annual number of published works pertaining to graphene modifications using click chemistry (see Fig. 1) signifies a bright future for this research field which can brings together the world of sugars and that of graphene chemistry. Of particular interest are Diels Alder click-reactions which become more and more popular. On the other hand, CuAAC click-reactions are very attractive since both azide- and alkyne- modifications of sugars are facile and have been long studied for other purposes (e.g. bioconjugation)."

Round 2

Reviewer 2 Report

The authors have improved the manuscript. It may be accepted for publication.